# Indigenous Social Policy and Social Inclusion in Taiwan

**Jiun-Hao Wang [1] and Szu-Yung Wang [2,*]**

[1] Department of Bio-Industry Communication and Development, National Taiwan University, Taipei 10617, Taiwan; wangjh@ntu.edu.tw

[2] Department of Agricultural Economics, National Taiwan University, Taipei 10617, Taiwan

* Correspondence: d03627001@ntu.edu.tw

**Abstract:** Social exclusion problems are inevitable in achieving social sustainability. Minorities or indigenous people encounter social exclusion from mainstream society in many countries. However, relatively little is known about the multiple disadvantages in different social welfare domains experienced by these indigenes. The objective of this study is to address indigenous social exclusion by focusing on their access to social welfare benefits. Data used in this study were drawn from the Social Change and Policy of Taiwanese Indigenous Peoples Survey, which included 2040 respondents. Logistic regression results revealed that, compared with their counterparts, the likelihood of being excluded from social welfare payments is higher for those who are plains indigenes, live outside of designated indigenous areas and participate less in local organizations. Besides varying the effects of ordinary explanatory variables on social exclusion across different exclusion models, this study further provides empirical evidence of the multidimensional disadvantages of indigenous peoples in receiving needed social welfare benefits.

**Keywords:** Sustainable Development Goals; social exclusion; social welfare; indigenous peoples; Taiwan

## 1. Introduction

Elimination of inequalities and social exclusion plays a critical role in achieving social sustainability. As mentioned in the Sustainable Development Goals (SDGs) by the United Nations, social inclusion is the key indicator of social sustainability. Moreover, implementing and achieving SDGs are meaningful for indigenous peoples to preserve a sustainable way of life, in which socio-ecological lifestyle is deeply tied with their territories, livelihoods and natural environment [1]. Despite social policy appearing to function as a redistributing instrument for reducing inequalities between the general population and vulnerable groups such as persons with disabilities and minorities, indigenous peoples suffer from lower income growth, educational attainment, medical expenses and even shorter life expectancy compared with non-indigenous people [2–4]. Several factors may underlie the differences between the ethnic group and its counterpart, such as lower educational attainment and higher unemployment rate (please refer to the Table A1 for detailed information) and lack of access to public assistance and medical care because of geographic remoteness, insufficient information and communication [2]. Problems of inequality and social exclusion are usually created and deepened by the economic growth pursued by the government [5]. Protecting the indigenous peoples' well-being and rights to social services and resources is therefore an important issue for social policy [6–8]. However, relatively little is known about the social exclusion of ethnic minorities, particularly in the field of the social welfare system itself.

The situation of social exclusions is neglected in some developed countries where indigenous peoples still suffer from racism, discrimination and unfriendly access to public services or basic

needs [9–11]. Similar problems also occur in East Asian countries such as Taiwan, which is reported as the third region in the world where racist behavior is frequently observed [2]. A large proportion of people face persistent social challenges and suffer from disadvantaged socio-economic status in Taiwan [12–15]. For example, Taiwanese aborigines have a shorter life expectancy, lower educational levels, lower family income, higher rates of single-parent families and unemployment than their non-indigenous counterparts [16–18]. An increasing body of literature indicates that racial discrimination, income inequality and decent work deficits are the major causes of indigenous exclusion and have hindered ethnic minorities in accessing health care, housing, education, employment and public services in the general social protection systems [2,13,19–21]. Indigenous exclusion from social benefits and resources refers not only to the availability but also the accessibility of social welfare programs to ethnic minority groups. Previous studies point out that the rural–urban disparity in accessing the provision of public goods and services is often pronounced. Apart from limited access to public services and resources in rural areas, a more serious exclusion affects indigenous peoples who live in remote highland areas, compared with the general rural residents [22]. In particular, mountain indigenes who are characterized by geographic remoteness, social isolation, racism and socio-economic disadvantages experience multiple problems related to social exclusion.

As recommended in the SDGs, to achieve social sustainability is to end poverty, provide better healthcare services and quality education and eliminate inequalities. The notion of social sustainability generally contains two parts in previous studies [23]. The first part is about creating and maintaining individual needs, physical and mental well-being and quality of life at an acceptable level for the entire ethnic group. The second part concerns the equality of policy and legislation and social justice and equity. In the context of this study, the social sustainability of indigenous peoples emphasizes protecting, encouraging and treating fairly ethnic groups and providing essential services or social welfare to meet their basic needs [21,24]. From the perspective of improving social sustainability, obtaining a better understanding of social exclusions of indigenous peoples may help us to mitigate the gaps between indigenous and non-indigenous peoples. However, since there is no available nationwide data for indigenous and non-indigenous peoples, we focus our research on the multidimensional social exclusion of indigenous peoples and we anticipate the implementation of more practical and effective policies. In the literature, many scholars adopt multidimensional social exclusion to quantify the situations of social exclusion, which is defined as a process of progressive dissociation by specific groups or individuals from participating in the general social system and institutions [6,8,21,24,25]. Therefore, indigenous social exclusion is suggested to be measured in several dimensions, for example, material resources, employment, education, medical and health care, community and personal safety [17,18]. Moreover, economic security, access to public resources and social benefits are the most important domains for multidimensional social exclusion [26]. However, relatively little is known about multiple exclusion in accessing social welfare programs that are experienced by indigenous peoples.

Previous studies indicate that social policy in various countries tends to provide monetary benefits rather than in-kind benefits because monetary benefits are more cost-effective and flexible. Specifically, monetary benefits are advocated as a means of empowering indigenous peoples in need of care because of the broader opportunities of choice [27]. Cash benefits of social welfare programs are considered a direct way to increase indigenous peoples' disposable income and major public support to secure the requisite materials for sustaining or improving their living conditions [28]. From the perspective of social policy, this study focuses on the association between welfare payment delivery and selects determinants to explore the multiple disadvantages faced by indigenous peoples in Taiwan. To fill this knowledge gap, the main objective of this study is to address indigenous social exclusion while paying special attention to indigenous peoples' access to social welfare benefits. In addition, this study employs several binaries and ordered logistic regression models to examine indigenous social exclusion in multiple welfare dimensions by using a national representative sample from the Taiwanese Indigenous Peoples Survey in 2007.

## 2. Literature Review

### 2.1. Overview of Indigenous Social Policy in Taiwan

According to an official statistical report, over half a million indigenous peoples accounted for 2.3% of the total population in Taiwan in 2008. The Council of Indigenous peoples (CIP) has officially recognized 14 tribes of indigenous peoples, mainly distributed in the east coast and southern, central and northern mountain areas in Taiwan. The government also designated 55 indigenous areas in 357 townships in 2002 [16]. The differences in ethnicity and geographic residence can be attributed to the recognition and division of indigenous peoples in Taiwan into two categories: plains tribes and mountain tribes [16,19]. The plains indigenous peoples mainly live in the rural villages of the plains region and are estimated to number between 150,000 and 200,000, whereas mountain tribes refer to indigenous peoples traditionally living in highland areas [14]. The mountain indigenous peoples are characterized by insufficient human capital, poverty and difference in culture because they are ethnic minorities, in addition to other factors such as geographic remoteness, spatial isolation and natural constraints. Moreover, Leigh and Gong [17] pinned down that cognitive gaps between indigenous and non-indigenous peoples lead to social exclusion. They have become among the most marginalized groups in the country.

Although most indigenous tribes still preserve their language, customs, living territory, tribal settlement and social structure, they face the negative impact of rapid modernization. Considering the specific needs of indigenous peoples and their developmental problems—such as remoteness, natural constraints of livelihood, lower life expectancy at birth and higher unemployment rate [16,17,19]—the CIP has been enforcing a number of national laws to protect the rights of indigenous peoples since the early 2000s. Some of these laws include the Regulations of Recognition of Indigenous Peoples, the Education Act for Indigenous Peoples and the Indigenous Peoples Basic Act (IPBA) [29]. In 2002, the Executive Yuan had launched a nationwide policy, the 'New Partnership Policy' to assist indigenous communities in identifying inherent land rights. In total, 30 mountainous and 25 plain indigenous traditional territories have been identified in Taiwan. This legislation that the IPBA passed in 2005 obligates the government to provide dedicated resources for indigenous groups in developing a system of self-governance, formulating policies to protect their basic rights and promoting the preservation and development of their traditions and cultures. The indigenous basic law aims to recognize, protect and promote the fundamental rights of indigenous peoples, as well as to enhance and ensure their sustainable socio-economic development [30]. In line with the spirit of the 2007 United Nations declaration on the rights of indigenous peoples, the CIP has been promoting social welfare programs for Taiwan's aborigines [30]. The IPBA Articles 24-28 stipulate that the government shall implement public health and health care policies to improve indigenous peoples' physical well-being and overall health.

The difference in indigenous cultures, needs and unique living environments drives the CIP to conduct a series of social welfare programs that differ from the general population to assist indigenous peoples with health care, housing, education and employment, as well as to safeguard their economic security. The indigenous social welfare measures include lowering the standards for living assistance subsidies, economic security, medical and health subsidies, housing and supportive programs, children and youth educational benefits, social insurance subsidies and other social work services [31]. Differing from previous studies [13,19], this study merely focuses on cash benefits for indigenous social welfare programs, rather than various living support services because social security payments are designed to secure material livelihood and maintain the basic financial needs of indigenous peoples.

According to the official regulations on social welfare benefits [31], the indigenous health allowance includes subsidies for National Health Insurance (NHI), medical transportation and nutrition. For example, the transportation subsidies for seeking medical advice are between NT $300 and NT $500 per time, depending on the resident's location, with a maximum of 10 times per year. For the housing subsidies, preferential loans or interest subsidies for building and repairing houses are offered to middle- and low-income indigenous families or those living in remote indigenous areas.

For emergency situations causing immediate risk to health, life or property damage, the indigenous peoples can also apply for several financial supports. The emergency aid includes funeral subsidies, aid for medical care, aid for disaster, unemployment assistance and emergency assistance for livelihood. Meanwhile, the education allowance for indigenous children and youth is comprised of the following: childcare, tuition and teaching subsidies for indigenous schoolchildren; education subsidies for low or mid-income indigenous students; lodging and meal allowance for indigenous students; and scholarships for education development and tuition subsidies for indigenous university students. The education subsidies for indigenous students vary from NT $2000 to NT $20,000 per semester, from attending primary school to college. Moreover, indigenous seniors aged over 55 years are eligible to receive the indigenes old-age welfare allowance (NT $3000 per month) if they do not participate in any social pension program.

### 2.2. Multidimensionality of Indigenous Social Exclusion

The term social exclusion originated in Europe; it was regarded as an extended notion of poverty with a focus on the social problems of specific individuals or groups being denied access to rights, opportunities, resources and public services that are typically available to most people in society [26]. Social exclusion refers to the processes and outcomes in which vulnerable groups are excluded from full engagement in civic life [19,32]. The systematic exclusion of certain communities of people includes lacking access to or suffering disadvantages in, housing, employment, healthcare, democratic participation and social activities. Previous studies have demonstrated that multiple exclusions are related to detaching specific groups or deprived areas from social relations and institutions, as well as preventing them from normal participation in society [25,26]. The multi-dimensionality of social exclusion emphasizes exclusion across multiple disadvantage domains, resulting in negative impacts on individuals' well-being and quality of life.

In a more sophisticated approach, three main domains have potential importance in social exclusion: resources, participation and quality of life [26]. The first and most important dimension is resource deprivation, which refers to the lack of access to economic and material goods, as well as public resources and services. Given that a lack of basic material necessities leads to an increased risk of poverty and social exclusion, income is the most obvious indicator of material and economic resources. Therefore, there are increasing policy studies addressing the importance of social security programs on lowering the risk of poverty and reducing economic hardship for the disadvantaged [33,34].

Indigenous peoples are ethnic minorities and often reside in remote rural areas; hence, they are the most disadvantaged groups both in developing and developed countries [9,11]. Take the Australia indigenous policy for example. The Steering Committee for the Review of Government Service Provision emphasizes that indigenous exclusion differs from other forms of social exclusion in Australia [35]. Therefore, the Australian government provides the Overcoming Indigenous Disadvantage (OID) framework to describe the multidimensional nature of indigenous social exclusion. The following domains and key indicators of the OID were selected: life expectancy, employment, disability and chronic disease, household income, early child development, education and training, healthy lives and economic participation.

Although awareness of the importance of the multi-dimensionality of social exclusion is increasing, no consistent conclusions can be drawn regarding the extent, typology and contents of measuring indigenous multiple exclusions. Considering theoretical integrity, existing welfare benefits and data availability, this study divides the multiple social exclusions into five domains: medical and health, housing, financial, education and old-age exclusion. First, the health exclusion is specified as inaccessibility to resources, services and opportunities to preserve or improve individual health status, such as NHI coverage, medical advice and a living allowance for recovery from the illness. According to European typology on social exclusion, housing exclusion defines people who live in insecure, inadequate or overcrowded places [36]. Financial exclusion refers to people who have difficulty in acquiring financial support from traditional markets or the public sector [26]. Meanwhile, education

exclusion is apparent among children and youth without adequate out-of-home care or with problems participating in regular school activities [19]. Moreover, old-age exclusion is related to neighborhood, community and social relations, mobility and the pension rights of the elderly [37].

In conclusion, social exclusion is a powerful disrupter of the development of social sustainability. The multi-dimensionality of social exclusion restricts the access of disadvantaged groups to resources and opportunities which would elicit social, economic and civic participation from them [38]. Although material resources do not simply equate to cash availability, income security is still the foundation of individual well-being and quality of life. Limited access to economic material resources is likely to lead to further exclusion from other life dimensions. The provision of cash benefits is recognized as a direct and effective instrument for removing the barriers to social inclusion throughout the social policy literature [39]. For overcoming social exclusion systematically, governments have been spending much effort on legislation and social welfare programs to reduce the inequitable exclusion of specific groups [35,38]. Therefore, this study concentrates on cash payments of different social welfare programs by measuring multi-dimensional social exclusion. The detailed operational definition and measurement of multidimensional social exclusion and independent variables are described below.

### 2.3. Determinants of Indigenous Social Exclusion

Social exclusion is the result of mutually reinforcing deprivations in the above-mentioned dimensions. Previous studies have identified several determinants that are associated with social exclusion [19,24]. Of these determinants, gender, age, ethnicity, education and religion—as well as specific status linked to income, health, employment, institutional system and residential location of the individual—may directly or indirectly affect social exclusion. The main reason for exclusion is the fact that indigenous peoples are part of an ethnic minority. Racism limits indigenous peoples' access to public resources and services; thus, eliminating this problem has become an important issue for social policy formation. Several arguments have addressed institutional discrimination that is blocking indigenous peoples from social system benefits because they are ethnic minority groups and often live in deprived rural areas with minimal choices [24,40].

The differences in research subjects, years studied, data sources, domains and measures of social exclusion have caused early studies to draw inconsistent conclusions about the relationship between selected determinants and social exclusion. For example, being older, having low income, having no private transport and not owning their accommodation signify those persons who are likely to be excluded from material resources [41,42]. Similarly, being female, in an ethnic minority group, living alone (having no partner, children or siblings) and being unemployed are risk factors for the exclusion of social relationships [43,44]. Although previous results show that possible determinants do not have the same effect on each exclusion dimension, the influence of socio-demographic characteristics, household features and environmental factors on social exclusion is well documented [17,19,26]. Relevant research findings reveal several key characteristics that are most likely related to multiple social exclusions experienced by indigenous peoples; they include age, gender, ethnicity, education, income, family type, employment, residence, mobility and membership of organizations.

In addition, while the social exclusion in rural areas is a widespread phenomenon, the spatial difference in social exclusion is also diverse among rural residents. For example, in the indigenous region, the specific environment of an area dominated by an indigenous ethnic majority is characterized not only by legal frameworks (e.g., traditional territory) but also by specific locality (e.g., distinct tribal culture and lifestyle) and natural circumstance (e.g., remoteness and geographic accessibility). These spatial factors interact with individual risk factors stemming from personal characteristics and produce different patterns and levels of exclusion. Furthermore, the role of indigeneity is crucial to understanding indigenous social exclusion. The World Bank indicated that indigeneity can be identified in particular geographical areas by the presence of specific characteristics, such as a close attachment to ancestral territories and natural resources, the presence of common cultural values and social structure

and primarily by subsistence-oriented production [45]. Enormous evidence suggests that indigeneity is a major risk factor in most exclusion domains of indigenous peoples [26,45].

Indigeneity refers not only to indigenous ethnicity but also to the embedded characteristics of geographic areas. First, ethnic indigeneity signifies those persons who self-identify themselves as a distinct ethnic group. Despite the diversity of cultures and living conditions of different indigenous peoples, they share common inherencies and group identities and differ from the dominant society [46,47]. Indigenous culture is tradition-oriented, involves a kinship-based tribal lifestyle and shapes indigenous institutions. Examples include distinct social norms, values and arrangements, a strong mutual support system and network and elder councils and chieftainship for organizing political, economic and social activities [17,48]. Such indigenous features manifest social exclusion and cause ethnic minorities to more likely face disadvantaged situations compared with Han people (known as Han Chinese) in the greater society [49].

In addition to ethnic differences, territorial indigeneity displays higher than average rates of geographic immobility, traditional indigenous residence, with hunter-gatherer or farming livelihood still being prevalent in remote rural areas. Previous studies demonstrate that remoteness and isolation of indigenous residence are likely to make the locals vulnerable to being disadvantaged in the development process [19,50]. Despite steady improvement in indigenous socio-economic status over the last two decades, they are still relatively worse off than other Taiwanese peoples in terms of education, employment, income and health. Moreover, they tend to live in more remote, deprived rural areas than other Taiwanese. As a result, indigenous peoples experience a substantially higher prevalence of social exclusion compared with non-indigenes. Inasmuch as our study focuses on the accessibility of social welfare delivery for indigenous peoples rather than general social exclusion issues, our empirical analysis might point towards divergences resulting from indigenous ethnic background and remote traditional residence in influencing social welfare exclusion.

## 3. Materials and Methods

### 3.1. Data

Data used in this study were drawn from the 2007 'Social Change and Policy of Taiwanese Indigenous Peoples Survey' (SCPTIP), conducted by the Institute of Ethnology in Academia Sinica [16]. This survey adopted a systematic sampling scheme which considers the indigenous population characteristics. After dropping missing values of crucial items, such as social welfare exclusion, our data included 2040 respondents in the empirical analysis. Using the nationwide representative indigenous survey, we constructed a set of measurements of multidimensional social exclusions for each individual, distinguishing the following five dimensions of a social welfare program—cash payments for medical and health, housing, financial, education and old-age allowances.

### 3.2. Measurements

To capture the multidimensionality of indigenous social exclusion, the dependent variable was measured by accessibility to different social welfare payments, in terms of exclusion from medical and health, housing, financial, education and old-age security benefits. With respect to social welfare exclusion, respondents of the SCPTIP survey were asked the following single choice question: 'If you have not received any of the above health or medical benefits, what is the main reason?' Optional answers include 'never heard,' 'no demand,' 'I want to apply but I do not know the application procedures' and 'others.' We could hardly identify whether or not respondents were eligible for claiming welfare payments because of the limitation of the survey content. We had to compromise this restriction of the existing questionnaire and classify those who chose an option except for 'no demand' as socially excluded from specific welfare payments in this study. In other words, those respondents who demanded social payments but did not receive social welfare are coded as dummy variables equal to 1 (1 = excluded) with respect to each dimension mentioned above.

Medical and health exclusion was measured as inaccessibility to NHI subsidies, transportation subsidies for seeking medical advice or medical allowance for indigenes (MEDICAL_EX). Housing exclusion was defined as respondents who demanded subsidies for housing and living arrangement which they did not receive (not well informed or having difficulties applying are included) or interest subsidies for building and repairing houses for indigenes (HOUSING_EX). Financial exclusion refers to persons who experience economic difficulties but did not acquire financial support from the public sector accordingly, such as aid for an emergency, unemployment assistance, low-interest loan, livelihood assistance or funeral subsidy (FINANCIAL_EX). Education exclusion is described as persons who have enrolled schoolchildren or students and could not access childcare and tuition subsidies, scholarships or other education allowances for indigenous students (EDUCATION_EX). The old-age exclusion was measured as senior family members aged over 55 years who could not receive any old-age welfare living allowances or other social security payments for senior indigenes (OLD-AGE_EX). Moreover, we summed up the total number excluded from the aforementioned indigenous social payments to represent the multidimensionality of social welfare exclusion (NUMBER_EX).

To examine the effects of the relevant determinants on multidimensional social exclusion, we employed ordinal regression analyses because of the ordinal scale of NUMBER_EX. Similarly, the other dependent variables of different kinds of social welfare exclusion were documented as binary variables (1 = excluded; 0 = not excluded). This study used several binary logistic regression models to examine the effects of selected factors on the likelihood of being excluded from different domains of social welfare payments while controlling for individual socio-demographic characteristics.

The socio-demographic variables included: gender (male = 1), age (in years), ethnicity (mountain indigenous person = 1), education level (e.g., three dummy variables of primary or lower, junior high school, senior high school and college or higher) and employment (employed = 1). Moreover, household features contained marital status (e.g., two dummy variables of married, single and others), average household income per month (in NT $ 10,000) and household size (in persons). Finally, the number of community organizations (in number), participation in local organizations (coded as 0 = *never*, 1 = *seldom*, 2 = *sometimes* and 3 = *usually*). Considering that the role of indigeneity is regarded as an important determinant of social exclusion [46,47], we also used indigenous ethnicity (e.g., mountain indigenous person = 1) and residence (e.g., lived in indigenous areas = 1) of respondents to identify the indigeneity and its effect on social welfare exclusion in further analysis. Worth mentioning, the subjective cognition of indigenous exclusion is included in the survey, 'Compared with Han people with the same conditions, do you think indigenous peoples are less likely to find a job?,' 'Compared with Han people with the same conditions, do you think indigenous peoples are less likely to be promoted?' Optional answers include 'yes,' 'no' and 'no comment' and over 45% of the respondents answered 'yes' in both questions. The detailed operational definition and descriptive statistics of multidimensional social welfare exclusion and explanatory variables are shown in Table 1.

**Table 1.** Variable definition and sample distribution (N = 2040).

| Variable | Definition | Mean (%) | SD |
|---|---|---|---|
| Dependent Variables (multidimensional social welfare exclusion) | | | |
| number_ex | Number of social exclusion dimensions if the respondent has demands for social welfare benefits but could not receive corresponding payments | 2.10 | 1.13 |
| medical_ex | Medical and health exclusion, if the respondent could not receive NHI subsidies, medical transportation subsidies or medical allowance for indigenes (=1) | 0.56 | - |
| housing_ex | Housing exclusion, if the respondent could not receive subsidies of housing or interest subsidies for building and repairing houses for indigenes (=1) | 0.54 | - |
| financial_ex | Financial exclusion, if the respondent encounters economic difficulty and could not receive aid for an emergency or financial assistance (=1) | 0.56 | - |
| education_ex | Educational exclusion if the respondent has any children enrolled in school and has not received childcare and tuition subsidies, scholarships and other education allowance (=1) | 0.30 | - |
| old-age_ex | Old-age exclusion, if the respondent has any family member aged over 55 and could not receive senior welfare living allowance (=1) | 0.17 | - |
| Socio-demographic variables | | | |
| male | If the respondent is male (=1) | 0.44 | 12.30 |
| age | Age of the respondent | 42.31 | 9.00 |
| mount_p | If the respondent is registered as mountainous indigenous people (=1) | 0.54 | - |
| primary | If the respondent has a primary school education or lower | 0.29 | - |
| junior | If the respondent's education level is junior high school | 0.22 | - |
| senior | If the respondent's education level is senior high school | 0.33 | - |
| college | If the respondent has a college level education or higher | 0.15 | - |
| employment | If the respondent currently has a job (=1) | 0.64 | - |
| Household features | | | |
| married | If the respondent is currently married (=1) | 0.62 | - |
| marri_other | If the respondent is divorced or a widower | 0.17 | - |
| single | If the respondent has never been married (=1) | 0.21 | - |
| num_household | Number of people in the household | 3.41 | 2.39 |
| inc | Average income per household in one month | 4.21 | 3.19 |
| Community/Regional characteristics | | | |
| org_community | Number of community organizations | 4.46 | 8.22 |
| org_partic | Frequency of the respondent attending local community organizations or associations | 1.00 | 1.09 |
| indig_area | If the respondent lives in an indigenous traditional territory (=1) | 0.51 | - |

## 4. Results

### 4.1. Descriptive Results

Table 1 presents the definition of variables, sample distributions of the multidimensional social exclusion and selected variables used in this study. Among the indigenous sample, 44% were male, with an average age of 42.31 years; 54% were registered as mountain indigenous persons; 33% of respondents had obtained senior high school, whereas only 15% of them had completed college-level education or higher; and 64% were employed. In addition, household features of the respondents showed that 62% were married, the average number of family members was 3.41 and the average household income was NT $42,100 per month. For the community or regional characteristics, on average, 4.46 local organizations exist in the community, respondents seldom participated in community organizations (mean = 1) and over half of the sample reported that they live in an indigenous traditional territory.

For the multi-dimensional social exclusion, overall, most of the respondents had experienced at least one dimension of social welfare exclusion. The percentage of the non-excluded group accounts for 8.3%. By contrast, only 34 respondents were unable to access all kinds of social welfare payments. Therefore, we combined all-excluded respondents into the 'over four-dimensional excluded group' which made up about 12%. On average, numerous respondents simultaneously suffered from more than two dimensions of social exclusion (mean = 2.1). In summary, for the specific dimensions of social welfare exclusion, over half of the total respondents were excluded from their needed medical and health benefits, housing subsidies and financial assistance. Medical and health exclusion and financial exclusion had the highest proportion (56%), while old-age exclusion has the lowest (17%).

### 4.2. Association Between Multidimensional Social Exclusion and Selected Variables

In order to investigate the extent to which socio-demographic variables, household features and community/regional characteristics may be associated with the multi-dimensionality of social exclusion, we divided the number of excluded social welfare benefits into five categories, including non-excluded and one- to four-dimension-excluded groups. For multiple group comparison, we used the Pearson's chi-squared test for independent variables which were categorical and one-way ANOVA for variables which were continuous.

Table 2 shows that the two-dimension-excluded group has the highest percentage (2_dimen_ex = 31.72%), whereas the non-excluded group has the lowest with 8.28% (not-excluded). In addition, those who had experienced exclusion from all kinds of social welfare benefits (4_dimen_ex) account for 11.86%. As expected, in comparison with the other groups, the non-excluded group was the youngest, wealthiest, better educated, with fewer household members and higher participation in local organizations. Conversely, respondents in the four-dimension-excluded group were the eldest, poorest and with more family members among all groups.

**Table 2.** Comparison of multi-dimensional social exclusion of indigenous peoples (N = 2040).

| | Not_Excluded N = 169 | | 1_Dimen_ex N = 461 | | 2_Dimen_Ex N = 647 | | 3_Dimen_Ex N = 521 | | 4_Dimen_Ex N = 242 | | $\chi^2$/F-Value |
|---|---|---|---|---|---|---|---|---|---|---|---|
| | Mean (%) | Sd. | Mean (%) | Sd. | Mean (%) | Sd. | Mean (%) | Sd. | Mean (%) | Sd. | |
| male | 0.51 | 0.50 | 0.44 | 0.49 | 0.45 | 0.50 | 0.43 | 0.50 | 0.36 | 0.48 | 10.92 ** |
| age | 39.42 | 13.04 | 42.57 | 13.24 | 42.40 | 12.30 | 42.40 | 11.81 | 43.36 | 11.53 | 2.81 ** |
| mount_p | 0.63 | 0.48 | 0.59 | 0.49 | 0.56 | 0.50 | 0.51 | 0.50 | 0.40 | 0.49 | 32.85 *** |
| primary | 0.20 | - | 0.29 | - | 0.29 | - | 0.29 | - | 0.35 | - | 47.71 *** |
| junior | 0.14 | - | 0.20 | - | 0.23 | - | 0.24 | - | 0.25 | - | - |
| senior | 0.38 | - | 0.32 | - | 0.33 | - | 0.34 | - | 0.31 | - | - |
| college | 0.28 | - | 0.18 | - | 0.14 | - | 0.12 | - | 0.09 | - | - |
| employment | 0.68 | 0.47 | 0.65 | 0.48 | 0.65 | 0.48 | 0.64 | 0.48 | 0.58 | 0.49 | 5.51 |
| married | 0.59 | - | 0.58 | - | 0.64 | - | 0.59 | - | 0.71 | - | 39.98 *** |
| marri_other | 0.12 | - | 0.16 | - | 0.16 | - | 0.21 | - | 0.17 | - | - |
| single | 0.30 | - | 0.27 | - | 0.20 | - | 0.20 | - | 0.11 | - | - |
| num_household | 3.18 | 2.74 | 3.21 | 2.51 | 3.43 | 2.39 | 3.43 | 2.27 | 3.87 | 2.08 | 3.44 *** |
| inc | 5.32 | 3.70 | 4.5 | 3.56 | 4.36 | 3.41 | 3.69 | 2.56 | 3.62 | 2.28 | 12.24 *** |
| org_community | 3.97 | 7.41 | 4.16 | 7.70 | 4.61 | 8.27 | 4.78 | 8.71 | 4.31 | 8.50 | 0.58 |
| org_partic | 1.24 | 1.14 | 1.18 | 1.10 | 1.04 | 1.12 | 0.77 | 1.02 | 0.84 | 1.03 | 12.76 *** |
| indig_area | 0.55 | 0.49 | 0.57 | 0.50 | 0.55 | 0.50 | 0.49 | 0.50 | 0.35 | 0.48 | 37.68 *** |

***, ** The significance at the 0.1% and 1%, respectively.

For socio-demographic characteristics, the results of the chi-squared and ANOVA tests indicated significant differences among the five groups in the aspects of gender, age, ethnicity and education ($p < 0.001$), except for employment. Compared with the not-excluded group, the respondents experienced more than four-dimensional exclusions (4_dimen_ex), characterized by several features, including female, older and less educated. Moreover, the household and regional heterogeneity in accessing social welfare payments were also significantly different among groups ($p < 0.001$). No statistical difference was found among different excluded groups with respect to the number

of community organizations, whereas frequency of attending local communities were significantly different among groups ($p < 0.001$). In the survey, major community organizations consists of religious organizations, cultural organizations, community development associations, agriculture production and marketing groups and tribal communities. As an institutional argument, those who frequently participate in community activities are less likely to be excluded. However, as a socio-economic status argument, it has not been proven that attending those activities may directly lead to be well-off. This result is consistent with previous studies that participation of these organizations may create opportunities for political participation, acquisition of skills, expansion of education, increase in social capital and enabling indigenous peoples to make more informed choices [2,51,52]. Other qualitative research also indicates that the effect of attending informal activities in local organizations is stronger than formal policy advocacy [53]. In general, those excluded from all social welfare payments (4_dimen_ex) were married, with more family members, lower household income and less participation in local organizations compared with their non-excluded counterparts (not-excluded). Perhaps the most interesting finding was the association between indigeneity and multi-dimensional social exclusion. The proportion of mountainous indigenes of each group significantly decreased ($\chi^2 = 32.85$, $p < 0.001$). From the not-excluded to the four-dimensional excluded groups, the percentages account for 63%, 59%, 56%, 51% and 40%, respectively. Residence also mattered for severity of indigenous social exclusion (chi-squared test = 37.68, $p < 0.001$). A large proportion of respondents live in an indigenous area and encounter fewer problems with receiving social welfare payment. Possible explanation is that over 70% of mountainous indigenous peoples live in their indigenous traditional territories and living in within the territories tends to have better connection. Moreover, attending local community activities (cultural activities are included) intensifies local social network and facilitates information sharing and processing, that is, those who frequently participate in cultural activities are less likely to be excluded [54]. The percentage of those living in indigenous areas is 55-57% for the excluded group with two or fewer dimensions.

### 4.3. Determinants of Multidimensional Social Exclusion of Indigenous Peoples

In order to obtain insights into selected variables associated with an indigenous respondent's likelihood of experiencing multi-dimensional social welfare exclusion, six logistic regression models were completed. Table 3 presents the estimations of the several logistic regression analyses, which include coefficients, standard errors, odds ratios (i.e., Exp (β)) and significance levels. We began our discussion of the results by looking at the findings of the statistical tests (bottom of Table 3). For the likelihood ratio test statistic of $M_{1-6}$, the Goodness-of-Fit value are 202.06, 50.16, 133.61, 283.23, 167.32 and 27.00, respectively, which were higher than the critical value at the 1% level ($p < 0.001$). Therefore, we rejected the null hypothesis that all slope coefficients are zero.

In Model 1 ($M_1$: Number_ex), an ordinal regression method was employed to examine how household features and community/regional characteristics affect the number of excluded social welfare benefits while controlling for socio-demographic variables. The respondents who were mountain indigenes, living in an indigenous area, with higher income and more active participation in local organizations were more likely to receive social welfare payment. Conversely, those who were junior or senior high school graduates, married and having more household members were more likely to experience more dimensions of social welfare exclusion, compared with their counterparts. For example, in the role of indigeneity, the mountain indigenes were about 0.68 times less likely to suffer from multidimensional social exclusions than those plains indigenes ($p < 0.001$). Moreover, those who lived in indigenous areas were 0.64 times more likely to receive social welfare benefits successfully than their counterparts ($p < 0.001$). These findings were plausible, inasmuch as Taiwan's government has promulgated several laws and regulations [30], as well as specific social welfare programs to protect the rights, way of life and economic security of indigenous peoples.

**Table 3.** Ordered/Binary logistic regression results of indigenous social exclusion (excluded vs. not_excluded).

| Parameter | M₁: Number_Ex [a] | | | M₂: Medical_Ex | | | M₃: Housing_Ex | | | M₄: Financial_Ex | | | M₅: Education_Ex | | | M₆: Old-Age_Ex | | |
|---|---|---|---|---|---|---|---|---|---|---|---|---|---|---|---|---|---|---|
| | β | Exp(β) | S.E. | β | Exp(β) | S.E. | B | Exp(β) | S.E. | B | Exp(β) | S.E. | β | Exp(β) | S.E. | β | Exp(β) | S.E. |
| male | −0.03 | 0.97 | 0.09 | −0.12 | 0.89 | 0.10 | 0.08 | 1.08 | 0.10 | −0.10 | 0.90 | 0.10 | −0.02 | 0.98 | 0.11 | 0.04 | 10.4 | 0.13 |
| age | −0.00 | 1.00 | 0.00 | 0.02 *** | 1.02 | 0.01 | −0.01 | 0.99 | 0.01 | 0.00 | 1.00 | 0.01 | −0.02 *** | 0.98 | 0.01 | 0.01 | 1.01 | 0.01 |
| mount_p | −0.38 *** | 0.68 | 0.08 | −0.22 ** | 0.80 | 0.09 | −0.30 ** | 0.74 | 0.10 | −0.02 | 0.98 | 0.10 | −0.33 *** | 0.72 | 0.11 | −0.40 *** | 0.67 | 0.12 |
| primary | 0.24 | 1.27 | 0.16 | 0.03 | 1.03 | 0.18 | 0.23 *** | 1.26 | 0.18 | 0.43 *** | 1.54 | 0.18 | −0.09 | 0.91 | 0.20 | 0.08 | 1.08 | 0.23 |
| junior | 0.31 * | 1.36 | 0.14 | 0.10 | 1.11 | 0.16 | −0.16 | 0.85 | 0.16 | 0.52 *** | 1.68 | 0.17 | 0.01 | 1.01 | 0.18 | 0.04 | 1.04 | 0.21 |
| senior | 0.23 * | 1.26 | 0.13 | 0.21 | 1.23 | 0.1 | −0.03 | 0.97 | 0.15 | 0.21 | 1.23 | 0.15 | 0.14 | 1.15 | 0.26 | 0.01 | 1.01 | 0.19 |
| (ref. = College) | | | | | | | | | | | | | | | | | | |
| employment | −0.05 | 0.95 | 0.09 | 0.00 | 1.00 | 0.10 | −0.01 | 0.99 | 0.10 | 0.03 | 1.03 | 0.11 | −0.19 * | 0.83 | 0.11 | 0.01 | 1.01 | 0.13 |
| marrie | 0.44 *** | 1.55 | 0.13 | −0.07 | 0.93 | 0.14 | 0.41 *** | 1.51 | 0.15 | 0.19 | 1.21 | 0.15 | 1.05 *** | 2.86 | 0.17 | −0.19 | 0.83 | 0.19 |
| marri_other | 0.47 *** | 1.60 | 0.16 | 0.12 ** | 1.13 | 0.18 | 0.55 *** | 1.73 | 0.19 | 0.20 | 1.22 | 0.19 | 0.59 *** | 1.80 | 0.22 | −0.16 | 0.85 | 0.24 |
| (ref. = single) | | | | | | | | | | | | | | | | | | |
| num_household | 0.09 *** | 1.09 | 0.02 | 0.02 | 1.02 | 0.02 | 0.02 *** | 1.02 | 0.02 | 0.03 | 1.03 | 0.02 | 0.15 *** | 1.16 | 0.02 | 0.06 ** | 1.06 | 0.02 |
| inc | −0.11 *** | 0.90 | 0.01 | −0.01 | 0.99 | 0.02 | −0.08 | 0.92 | 0.02 | −0.24 *** | 0.79 | 0.02 | −0.01 | 0.99 | 0.02 | 0.01 | 1.01 | 0.02 |
| org_community | −0.00 | 1.00 | 0.00 | 0.01 | 1.01 | 0.01 | 0.01 | 1.01 | 0.01 | −0.01 | 0.99 | 0.01 | −0.01 | 0.99 | 0.01 | −0.01 * | 0.99 | 0.01 |
| org_partic | −0.22 *** | 0.80 | 0.04 | −0.10 ** | 0.90 | 0.04 | −0.18 | 0.84 | 0.04 | −0.12 *** | 0.89 | 0.05 | −0.16 ** | 0.85 | 0.05 | −0.04 | 0.96 | 0.06 |
| indig_area | −0.44 *** | 0.64 | 0.09 | −0.30 *** | 0.74 | 0.10 | −0.65 *** | 0.52 | 0.10 | 0.05 | 1.05 | 0.11 | −0.24 ** | 0.79 | 0.11 | −0.22 * | 0.80 | 0.13 |
| Intercept_1 | 2.94 *** | 18.92 | 0.24 | −0.17 | 0.84 | 0.35 | 1.39 *** | 4.01 | 0.26 | 0.81 ** | 2.25 | 0.27 | −0.65 ** | 0.52 | 0.28 | −1.77 *** | 0.17 | 0.34 |
| Intercept_2 | 1.67 *** | 5.31 | | | | | | | | | | | | | | | | |
| Intercept_3 | 3.09 *** | 21.98 | | | | | | | | | | | | | | | | |
| Intercept_4 | 4.67 *** | 106.70 | | | | | | | | | | | | | | | | |
| Likelihood Ratio | 202.06 | | | 50.16 | | | 133.61 | | | 283.23 | | | 167.32 | | | 27.00 | | |
| Pseudo-R² | 0.03 | | | 0.02 | | | 0.05 | | | 0.10 | | | 0.08 | | | 0.02 | | |

[a] Reference group is not_excluded samples (Number_ex=0); ***, **, * The significance at the 0.1%, 1% and 5 % level, respectively.

Taking the non-excluded as the reference group for all models, we adopted identical variables for the following binary logistic regression models: medical and health allowance exclusion, housing subsidy exclusion, financial aid exclusion, education subsidy exclusion and old-age security payment exclusion. Similar patterns were found across $M_{2-6}$ estimations but significant differences existed. In Model 2 ($M_2$: Medical_ex), age and divorced or widowed status (single is the reference group) were positively associated with an increase in the proportional odds of being excluded from medical and health allowance. The respondents who were mountain people, participate more in community organizations and live in an indigenous area are more likely to access medical and health payment, compared with their counterparts, by 20%, 10% and 26%, respectively.

For housing exclusion ($M_3$), our results showed that those who are plains indigenes, with primary school or lower education, married, having more family members and living outside of indigenous areas, are more likely to be excluded from housing subsidy, compared with their counterparts. In the financial exclusion model ($M_4$), people who obtained junior high school or lower education are less likely to receive financial aid for an emergency compared with those who have college or higher education; in addition, the respondents who have higher household income and often attend community organizations are more likely to be included in financial aid programs. However, no statistically significant association was found between indigeneity and financial aid exclusion.

The exclusion model of education subsidy ($M_5$) indicated that those who are elderly, mountain indigene, employed, single, intensely participative in local organizations and settled in an indigenous area have a higher probability of receiving an education subsidy for their children's schooling. In the old-age exclusion model ($M_6$), mountain ethnicity, having more community organizations and indigenous residence are negatively associated with being excluded from old-age welfare programs; that is, these respondents are more likely to receive old-age security payment, compared with their counterparts.

Although the effects of explanatory variables on social welfare exclusion vary across different models, the direction and significance of relevant determinants remained largely unchanged. Our study yielded several interesting findings. For instance, we observed the statistically significant effect of indigeneity on social exclusion in most social welfare dimensions, except for the financial aid model. It was evident that the mountain indigenes and residence in indigenous areas significantly contribute to reducing the likelihood of being excluded from social welfare benefits. The probable explanation was that mountain ethnicity and indigenous region make it convenient for social agencies to approach and deliver welfare payments to their target groups. In addition, the mountain indigenous peoples living in their traditional territory have a better chance to maintain their family clans, ethnic identity, traditional lifestyle and mutual support systems in local communities; thus, they can enhance their social solidarity and mobilize social welfare delivery. Indigeneity did not help to ameliorate social welfare exclusion in terms of financial aid for economic hardship because of the remoteness of indigenous areas and isolated settlement of mountain indigenes [51,52,54]. Furthermore, our findings pointed out that the higher the participation frequency in local organizations, the lesser is the probability of being excluded from social welfare payments. The number of organizations in a community also played an insignificant role in the social welfare exclusion of indigenous peoples.

## 5. Conclusions

To achieve social sustainability, we inevitably need to face the challenges of social exclusions. The problem of social exclusion is usually tied to many aspects such as poverty, education, medical care and inequalities, which could not be discussed independently. This study focuses on the multi-dimensional social exclusion of indigenous peoples, regarded as an ethnic minority, who are vulnerable to external influences, discrimination and marginalization by the majority in Taiwan [16,17]. The primary purpose of this study is to provide empirical evidence accumulated from exploring the multi-dimensional social exclusion of indigenous peoples. Using the nationwide survey data in Taiwan, our empirical findings reveal some interesting findings with policy implications. First, the reported

findings show that over half of the respondents (54.32%) are excluded from two or three kinds of welfare payments. Among different social welfare exclusions, inaccessibility to medical and health allowances, housing subsidies and financial aid for an emergency are the severest forms of indigenous social exclusion, as defined in this analysis. Less than 10% of respondents have successfully received social welfare benefits to meet their needs. This end result of multiple exclusion points to the evidence that ignoring the accessibility to social welfare payment may result in misleading indigenous policy inferences.

Second, a statistically significant association between indigeneity and multi-dimensional social exclusion is found across different models. Interestingly, both ethnic and regional indigeneities positively contribute to reducing the risk of being excluded from social welfare benefits. Our results indicate that mountain indigenes living in an indigenous area are less likely to experience exclusion in most dimensions of social welfare benefits, except for the financial exclusion model. While these results may not be astounding and are consistent with prior expectations, they point to the importance of accessing rights to social welfare systems to alleviate indigenous social exclusion problems [14,17,54]. Consistent with the findings of previous studies, mountainous indigenous who live in tribal areas periodically gather to hold activities such as ceremonies, reunions or rituals. Those activities promote the flow of crucial information delivery and exchange tribes and the experience of applying social welfare may also be included [54]. Our results are also consistent with the findings of the previous study [55] that determined indigenous traditional territory as a key factor in the positive association with reductions in social exclusion.

From the perspective of social sustainability, our findings have significant implications regarding the multidimensionality of social exclusion and indigenous social policy. First, for the policy-makers, the higher levels of social exclusion with indigenous features suggest that existing social welfare programs are still unable to prevent indigenous peoples from becoming socially excluded. Besides, fiscal policy cost is another priority issue that is based on knowing well the influences of those welfare and benefits. Therefore, our findings suggest that indigenous policy should put more effort into maintaining and strengthening territorial and ethnic indigeneity, as these have an indirect but positive influence on functioning indigenous welfare programs. In particular, this approach supports many of the arguments for promoting traditional territory, tribal lifestyle and ethnic solidarity among indigenous peoples in relation to indigenous policy implications. Second, indigenous policy-makers should regard multi-dimensional exclusion as a policy priority. Our results suggest that further measures should focus on the most excluded domains, such as medical and health exclusion. The government can encourage the local social agencies and indigenous organizations to develop simplified and transparent administration of social welfare registration for indigenous peoples. Considering that different welfare programs and single-focus policy administration have little contact with one another, although not directly related to our study, the government could also transcend bureaucratic barriers by integrating discrete welfare-related authorities, such as indigenous administration, social affairs, health, education and economic affairs departments and provide one-stop service to help indigenous peoples gain access to social welfare benefits. Third, we suggest that official authorities deliver one set of cash benefits and standardize the eligibilities of applicants to increase the accessibility and take-up rates. The rationale for bundled cash benefit is based on our findings that a significant interaction effect exists between different kinds of cash benefit (please refer to Table A2 for more details). Meanwhile, the current application process and eligibilities are disparate and confusing, which may defer indigenous peoples in need of immediate assistance. The practices mentioned above are anticipated to offer a more direct and effective policy that can be implemented to alleviate social exclusions and take us one step closer to achieving social sustainability.

Although some inspiring findings are revealed in this study, some limitations remain. First of all, the 2007 SCPTIP is the only available dataset related to the social welfare of indigenous peoples. Many gaps exist between the present situations and the social benefits in 2007. For instance, the CIP has implemented the 'Social Security Development for Indigenous Peoples Project' in 2009 [29].

Moreover, in particular, the introduction of the National Pension Insurance in 2008 has offered universal pension benefits to indigenous older people, replacing the old-age indigenes' welfare living allowance. Such reforms have resulted in changes in eligibility and requirements for old age payments; therefore, it is hard to distinguish qualified indigenous peoples after the reform through the existing surveys. Since we could hardly identify whether the respondents were entitled to the social welfare, those people who need cash benefits but were not eligible were included. On the other side, some people chose not to claim their benefits because of insufficient amount of payment and other reasons. Fortunately, these people can be clearly distinguished from the survey and their proportion is relatively low (1.52%). Consequently, the empirical results of social exclusion may be slightly overestimated. Another limitation of this study is that we can hardly differentiate the types of exclusion because of lack of access (supply-side problem) or lack of demand (demand-side problem). We suggest future research to consider or adopt qualitative fieldwork to overcome this issue. Besides, in alignment with other studies using nationally representative data, the information concerning the accurate measurement of social exclusion is limited. Consequently, our study will be more robust if further information is available about the main reasons why the respondents did not apply for or receive, social welfare payments.

**Author Contributions:** Conceptualization, S.-Y.W. and J.-H.W.; Methodology, J.-H.W.; Software, J.-H.W.; Formal Analysis, J.-H.W.; Resources, J.-H.W.; Data Curation, J.-H.W.; Writing—Original Draft Preparation, J.-H.W.; Writing—Review & Editing, S.-Y.W. and J.-H.W.

**Funding:** The authors acknowledge financial support from the Chung-Hua Institution for Economic Research and National Science Council (Grant No. NSC 100-2621-M-002-017). The results and their interpretation are those of the authors, and are not those of the Chung-Hua Institution for Economic Research and National Science Council in Taiwan.

**Conflicts of Interest:** The authors declare no conflict of interest.

## Appendix A

**Table A1.** Social indicators comparison between indigenous and non-indigenous people.

| Educational Attainment | Indigenous Peoples | Non-Indigenous People | Differences (Gap) |
|---|---|---|---|
| Elementary school and under* | 27.3% (48.3%) | 18.1% (30.5%) | 9.2% (17.8%) |
| Junior high school* | 21.6% (75.1%) | 12.6% (63.1%) | 9.0% (12%) |
| Senior high school* | 35.9% (67.3%) | 31.9% (63.9%) | 4.0% (3.4%) |
| Vocational school* | 6.5% (72.7%) | 12.8% (76.8%) | −6.3% (−4.1%) |
| University and above* | 8.7% (52.3%) | 24.6% (62%) | −15.9% (−9.7%) |
| Unemployment rate | | | |
| Male | 4.10% | 3.90% | 0.20% |
| Female | 4.60% | 3.70% | 0.90% |
| Household income and expenditure (NT$**) | | | |
| Disposable income | 458,000 | 888,000 | −430,000 |
| Education expenditure | 30,672 | 144,011 | −113,339 |
| Medical expenditure | 33,101 | 101,969 | −68,868 |
| House expenditure | 58,761 | 171,482 | −112,721 |
| Expectancy life | | | |
| Male | 63.9 | 74.5 | −10.6 |
| Female | 73.1 | 80.8 | −7.7 |

Statistics are collected from Annual Report of Indigenous peoples in Taiwan, Council of Indigenous People [56].
* Labor force participation rate is shown in parentheses; ** All of the monetary terms are deflated to the 2007 level.

**Table A2.** Pearson correlation of multiple social welfare exclusion.

|  | Medical_Ex | Housing_Ex | Financial_Ex | Education_Ex | Old-Age_Ex |
|---|---|---|---|---|---|
| medical_ex | 1.000 |  |  |  |  |
| housing_ex | 0.168 *** | 1.000 |  |  |  |
| financial_ex | 0.077 *** | 0.117 *** | 1.000 |  |  |
| education_ex | 0.037 * | 0.058 *** | −0.002 | 1.000 |  |
| old-age_ex | −0.007 *** | −0.018 ** | 0.041 ** | 0.120 | 1.000 |

***, **, * The significance at the 0.1%, 1% and 10% level, respectively.

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
