# Peer review of "Indigenous Social Policy and Social Inclusion in Taiwan"

_sustainability, doi:10.3390/su11123458_

Round 1

Reviewer 1 Report

I think overall this is an interesting paper but needs two bigger changes:

 - a discussion that engages a bit more with the findings, their fit with other existing research on the topic, and their policy implications

 - language editing (there are a lot of fragmented sentences and other grammatical mistakes throughout the text at the moment)

More minor comments I made while reading: 

-        Keep mentioning that social exclusion is a key factor in social sustainability (or lack thereof) – but need to define what social sustainability is

-        Refer to the United Nations ‘Leaving No One Behind’ report from 2016 on social exclusion

-        Why are only indigenous people observed in the study? I think the study could provide more insight into the issues of exclusion if also non-indigenous people were included – was there no appropriate data available? You touch on this a little bit in the conclusion but I think you need to discuss earlier the limitations of the study from this perspective.

-        Needs to be clarified if all people counted as ‘excluded’ are in fact actually entitled to the welfare programmes – and if not, as it seemed to me at some point from the text, then you need more justification/clarification whether this can actually be done and how it can affect (bias?) the results

-        Need to make clear that some ‘social exclusion’ may not be that at all – but rather people choosing not to claim the benefits to which they may be entitled for other reasons

Author Response

Dear reviewer, we thank you for your constructive comments and suggestions. We have incorporated many of your suggestions into the revised manuscript where appropriate. By making these changes, the quality of this paper is significantly improved. Please refer to the file attached, we address each of your specific comment in detail.

Reviewer 2 Report

Indigenous Social Policy and Social Inclusion in Taiwan

This is a very strong paper, and very topical. It nicely brings the indigenous development agenda to within the broader SDG agenda. This is especially important as the paper – and main thesis of the paper, which is the systemic exclusion of indigenous communities and people from development – strengthens the claim the SDGs are not just for poor countries, but also relatively rich ones like Australia, Mexico, Canada, and Taiwan.

The paper’s goals are actually quite simple: to demonstrate that indigenous peoples are excluded from social policy benefits (specifically cash benefits). The paper would be strengthened if the authors showed how indigenous people in Taiwan were more excluded than non-indigenous and specifically by how much. To be sure, one of the drawbacks of the paper is that the data used in the quantitative analysis is quite old (12 years old, from 2007 survey) and quite limited in scope (to just indigenous people). In the end the paper provides some empirical evidence of indigenous exclusion. The thesis is intuitive and the findings are not particularly surprising. But as far as empirical verification of what we might expect, the paper is more than adequate. In fact, it’s quite needed.

A few comments intended to help improve the paper and strengthen its potential impact.

1.       The paper establishes empirically that indigenous people in Taiwan are excluded. It speculates as to why this may be the case, but there is little explication as to why we see the patterns we do. There is little causal analysis aside from more general statements about geographical distance, racism, etc. In the absence of more recent and broader quant data, perhaps the authors could share some qualitative insights into the precise mechanisms as to why we see exclusion. For instance, what is it about “racism” that creates exclusion; what is it about geographical distance that leads to greater exclusion? We can only speculate.

2.       That said, the paper makes clear that being indigenous itself is not entirely consequential, and thus racism or ethnic discrimination are not the only variables at play. In fact, exclusion within indigenous groups is quite variable. Again, this is not surprising (i.e. the more educated, the more well-off are less excluded), but some explanation as to why would be helpful. To what extent is exclusion due to location variables versus cultural variables?

3.       The paper contends that mountainous indigenous communities are less excluded. This seems counter-intuitive if we consider the topographical and (lack of) density challenges in mountainous area. Can the author(s) explain why mountain-based indigenous communities are less excluded?

4.       The quant analysis also finds that those who participate in groups are less likely to be excluded. This is a finding consistent with findings in other cases. But what’s causal reasoning behind this? It’s not clear if associational life in the Taiwan case increases inclusion because (i) greater participation contributes to better governance (i.e. an institutional argument) or (ii) those in groups are also more likely to be more well-off (i.e. an SES argument)?

5.       I agree with the author(s)’ policy recommendation that potentially one way to increase accessibility and take-up rates among indigenous people would increase if the cash benefits were bundled and delivered together (i.e. if the government can deliver one set of cash benefits, why not more of them if they were bundled together). This raises an important question that the paper could shed more light on: is there any interaction effect between the different kinds of cash benefits? Is someone more likely to have access to benefit X if they access benefit Y, or perhaps if they access Z instead?

6.       The key challenge for this paper is making the distinction – both conceptually and empirically – between exclusion due to lack of access (i.e. a supply side problem) or lack of demand / awareness / education (demand-side problem). More qualitative work would be helpful in understanding WHY certain groups are excluded.

Author Response

(The authors gave the same response as above.)

Round 2

Reviewer 1 Report

All my original comments had been addressed by the authors.

Author Response

Dear reviewer, 

It is much appreciated for your fast response and constructive comment. 

We have carefully revised and edited the manuscript word by word and have sent to a professional English editing service for further revision again. We sincerely hope the revisions will fulfill the requirements of this journal. Please refer to the attachment for more details.

Again, we thank you for your kindness and valuable time. 

Kind regards,

Authors
